# Creating enzymes and self-sufficient cells for biosynthesis of the non-natural cofactor nicotinamide cytosine dinucleotide

Xueying Wang[1,2], Yanbin Feng [1], Xiaojia Guo[1,2], Qian Wang[1,2], Siyang Ning[1,2], Qing Li[1,3], Junting Wang[1,3], Lei Wang[1] & Zongbao K. Zhao [1,2,4 ✉]

Nicotinamide adenine dinucleotide (NAD) and its reduced form are indispensable cofactors in life. Diverse NAD mimics have been developed for applications in chemical and biological sciences. Nicotinamide cytosine dinucleotide (NCD) has emerged as a non-natural cofactor to mediate redox transformations, while cells are fed with chemically synthesized NCD. Here, we create NCD synthetase (NcdS) by reprograming the substrate binding pockets of nicotinic acid mononucleotide (NaMN) adenylyltransferase to favor cytidine triphosphate and nicotinamide mononucleotide over their regular substrates ATP and NaMN, respectively. Overexpression of NcdS alone in the model host *Escherichia coli* facilitated intracellular production of NCD, and higher NCD levels up to 5.0 mM were achieved upon further pathway regulation. Finally, the non-natural cofactor self-sufficiency was confirmed by mediating an NCD-linked metabolic circuit to convert L-malate into D-lactate. NcdS together with NCD-linked enzymes offer unique tools and opportunities for intriguing studies in chemical biology and synthetic biology.

[1] Laboratory of Biotechnology, Dalian Institute of Chemical Physics, CAS, Dalian, PR China. [2] Dalian Key Laboratory of Energy Biotechnology, Dalian Institute of Chemical Physics, CAS, Dalian, PR China. [3] University of Chinese Academy of Sciences, Beijing, PR China. [4] State Key Laboratory of Catalysis, Dalian Institute of Chemical Physics, CAS, Dalian, PR China. ✉email: zhaozb@dicp.ac.cn

**N**icotinamide adenine dinucleotide (NAD, Fig. 1) and its reduced form NADH are indispensable cofactors that participate in diverse redox biochemistry and are also involved in other nonredox processes[1]. Tremendous efforts have been devoted to manipulate cellular NAD(H) level and related metabolites for advanced metabolic engineering and therapeutic interventions[2]. However, the success rates are compromised because NAD(H) level fluctuation can lead to global yet difficult-to-predict biological responses[3]. A few NAD mimics have been shown to mediate redox chemistry intracellularly, albeit their applications are limited to enzymes that retain residual activities with those mimics[4]. Specifically, N$^{TZ}$AD that contained iso-thiazolo[4,3-d]pyrimidine moiety was synthesized enzymatically and accepted as redox cofactor by wild-type alcohol dehydrogenase and glucose-6-phosphate dehydrogenase form *Saccharomyces cerevisiae*[5,6]. A stable NAD mimic, 4′-thioribose NAD (S-NAD), was prepared via chemoenzymatic process and was active with bovine glutamate dehydrogenase[7]. As abiotic precursors and multistep organic synthesis are required to prepare N$^{TZ}$AD and S-NAD, it remains a challenge to use them as redox cofactors for metabolic engineering. Interestingly, nicotinamide mononucleotide (NMN), an endogenous metabolite and a precursor for NAD biosynthesis, was demonstrated as an effective cofactor to mediate redox chemistry in *Escherichia coli* whole cells[8,9].

In our early study, an NAD analog, nicotinamide cytosine dinucleotide (NCD, Fig. 1), was chemically synthesized and malic enzyme (Mae) mutants were obtained to use NCD for oxidative decarboxylation of L-malate[10]. We then engineered a number of NAD-dependent enzymes, including phosphite dehydrogenase, formate dehydrogenase, and D-lactate dehydrogenase to favor NCD[11–14], and found wild-type nucleotide transporter NDT2 form *Arabidopsis thaliana* and NTT4 originated from *Protochlamydia amoebophila* UWE25 were efficient to shuttle NCD into *E. coli* cells[15]. By using these genetic parts, we have successfully devised metabolic circuits for pathway-selective chemical energy transfer, suggesting NCD potentially as a non-natural cofactor to establish more orthogonal redox chemistry. Nonetheless, the expression of nucleotide transporter led to reduced cell growth and feeding NCD was cost-prohibitive to scale up cell cultures.

To facilitate NCD-linked redox chemistry in vivo, it is essential to realize NCD biosynthesis. In bacteria, nicotinic acid mononucleotide (NaMN) adenylyltransferase (NaMNAT) catalyzes the condensation of ATP and NaMN to form nicotinic acid adenine dinucleotide (NaAD) (Fig. 1a), which is the key step for de novo NAD biosynthesis[16]. Conceptually, if the binding pockets of ATP and NaMN of NaMNAT are redesigned to favor cytidine triphosphate (CTP) and NMN, respectively (Fig. 1b), the engineered enzyme is expected to make NCD following a similar condensation mechanism, and can be designated as NCD synthetase (NcdS). As both CTP and NMN are endogenous metabolites, no auxiliary nutrients are required to synthesize NCD by those NcdS-empowered cells. However, it is intimidating to engineer a two-substrate enzyme for active variants specific with two new substrates. On one hand, simultaneous mutating key residues within two-substrate-binding pockets will lead to large libraries beyond our screening capacity. On the other hand, engineered molecular interactions favoring one substrate may have unanticipated effects on those for the other, and vice versa.

Here, we report the results of creation and application of NcdS by reprogramming the substrate preference of NadD, the gene *nadD* (Gene ID: 945248) product for NaMNAT in *E. coli*[17]. We employ semirational strategies to identify NadD mutants with strong CTP preference over ATP, followed by incorporating dedicated mutations previously devised for NMN preference[18], leading to the expected synthetase. When NcdS alone was expressed in *E. coli*, cells indeed produced NCD, and up to 5.0 mM NCD was formed when the biosynthesis of the precursors was enhanced. We further demonstrate that NCD self-sufficient cells were able to support NCD-linked metabolic circuit for orthogonal redox chemistry. This study provides NcdS to enable NCD biosynthesis and should promote further applications of NCD as a non-natural cofactor in chemical biology and synthetic biology.

## Results

**Creation of NcdS.** Our strategy to create NcdS was to change the binding pockets for ATP and NaMN of wild-type NadD separately, and then integrate those mutations. This strategy was proposed because the crystal structure of NadD indicates that the ATP-binding pocket and NaMN-binding pocket are spatially isolated[19]. As NadD has residual activity with NMN, we decided to evolve its ATP-binding pocket for CTP first. Thus, an in vitro assay was designed for those CTP-favoring hits in the presence of excess NMN (Fig. 2a). Here, Mae* is an NCD-specific malic enzyme[10] that enables a coupled colorimetric assay to identify

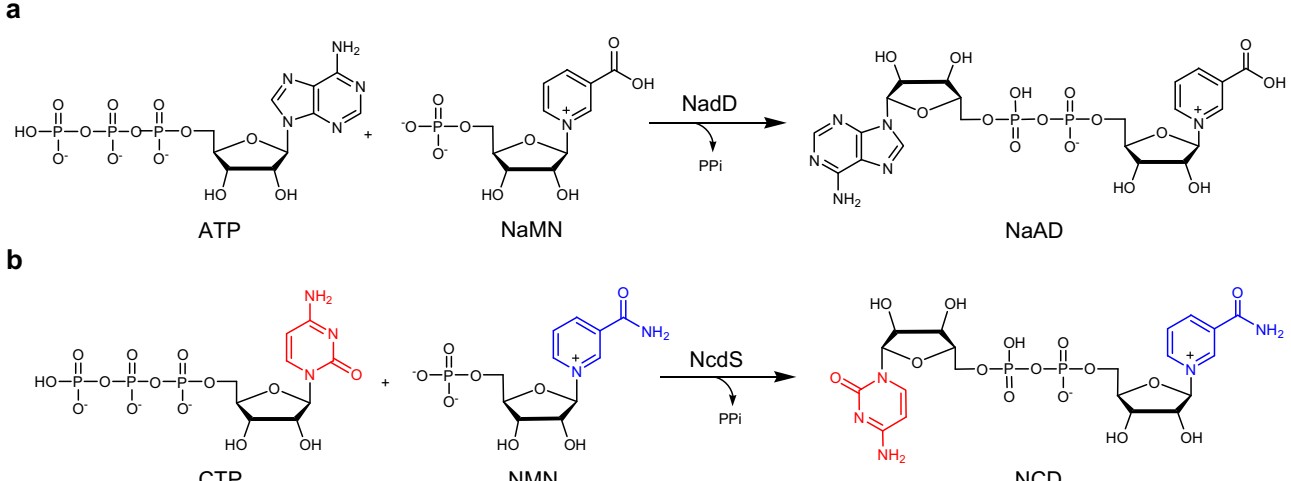

**Fig. 1 Structures and enzymes for cofactor biosynthesis. a** Native NadD catalyzed synthesis of NaAD from ATP and NaMN. **b** The proposed NcdS to catalyze the condensation of CTP and NMN into NCD.

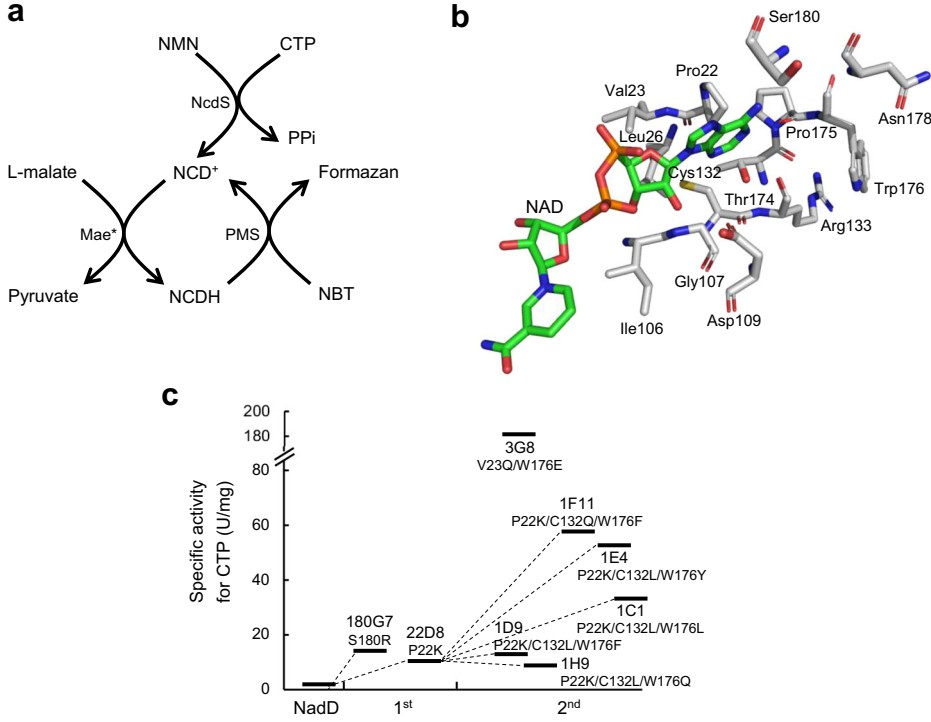

**Fig. 2 Directed evolution of NadD for NCD biosynthesis. a** The principle of coupled colorimetric assay for NCD formation. The expected NCD synthesis activity converts CTP and NMN into NCD, and an NCD-dependent malic enzyme mutant Mae* is used to convert NCD into NCDH, which reduces NBT into formazan in the presence of PMS. **b** The ATP-binding domain of *E. coli* NadD (PDB ID: 1K4M). NAD and targeted amino acids in NadD are shown in stick mode. Blue, nitrogen; red, oxygen; orange, phosphorus; yellow, sulfur; gray, carbon in NadD; green, carbon in NAD. **c** The evolutionary tree of NadD for CTP preference in two rounds of mutagenesis and screening.

mutants capable of synthesizing NCD in the presence of L-malate, nitroblue tetrazolium (NBT), and phenazine methosulfate (PMS). For the first round of directed evolution, we targeted residues locating within 8 Å to the adenine moiety (Fig. 2b), namely, Pro22, Val23, Ile105, Gly107, Asp109, Cys132, Arg133, Thr174, Pro175, Trp176, Asn178, and Ser180, and constructed single-site saturation mutation libraries. Screening results showed that the mutants P22K and S180R exhibited measurable CTP-utilization activity. We constructed the double mutant P22K/S180R, unfortunately, it showed reduced CTP-dependent activity (Supplementary Fig. 1). It seemed that the presence of bulky and charged side chains at position P22 and S180 simultaneously was antagonistic in terms of CTP recognition.

Next, we constructed multiple-site mutation libraries to cover more structural diversity. Considering that CTP is relatively smaller-sized than ATP, it is desirable to reduce the nucleoside triphosphate (NTP) binding pocket. Thus, we employed the triple-code saturation mutagenesis strategy[20,21] to introduce amino acid with a large-side chain, including Arg, Gln, Glu, Leu, Phe, and Tyr. Based on the codon usage and bias, degeneracy base group with equimolar ratio of ckc, twt, and saa was used as primers to cover those amino acids at each mutation site. Seven mutation libraries were constructed, namely, P22K-C132/W176, C132/W176, C132/S180, P22/V23/W176, V23/L26/W176, C132/A172/T174, and C132/T174/S180. Screening results showed that 1E4 (NadD-P22K/C132L/W176Y), 1F11 (NadD-P22K/C132Q/W176F), and 3G8 (NadD-C132Q/W176E) exhibited relatively high specific activity with CTP, whereas 3G8 had comparable specific activity with both ATP and CTP (Supplementary Fig. 2). The evolutionary tree of those mutants with improved activity to CTP was summarized in Fig. 2c. These active variants suggested that the P22K mutation attenuated ATP utilization, while large side-chain residues at position 132 and 176 promoted CTP utilization.

To ensure NCD biosynthesis, the NaMN-binding pocket was changed to favor NMN. In a previously study, a double-site mutant NadD-Y84V/Y118D was found with substantially improved preference to NMN for amidated NAD biosynthesis in *E. coli*[18]. By using the restriction-free (RF) cloning strategy, we introduced both Y84V and Y118D mutations into the CTP-preferring mutants 1E4 and 1F11 (Supplementary Fig. 3), and designated the final variant as NcdS-2 and NcdS-3, respectively. Both NcdS-2 and NcdS-3 are NadD variants harboring 5 mutations.

**Kinetic analysis.** The catalytic properties of NcdS-2 and NcdS-3 were estimated with different substrates. We purified both proteins to near homogeneity from *E. coli* DH10B cells harboring the corresponding expression plasmids. As NcdS held two pockets for their respective substrates, we determined the kinetics of synthesis of all four potential products, namely, NCD, nicotinic acid cytosine dinucleotide (NaCD), NAD, and NaAD, according to coupled colorimetric assays (Supplementary Fig. 4). Briefly, activities for synthesis of NCD and NaCD were estimated by using Mae* to generate their corresponding reduced forms. To estimate NAD synthesis activity, commercially available ADH II was used to reduce NAD to NADH, while NaAD synthesis activity was estimated by further recruiting NadE from *E. coli* to amidate NaAD to NAD.

The kinetic data are shown in Table 1. When NaMN is fixed, wild-type NadD using ATP for NaAD synthesis is over 57000-fold more efficient than using CTP for NaCD synthesis; while when ATP fixed, NadD using NaMN for NaAD synthesis is 246-fold more efficient than using NMN for NAD synthesis. These data confirmed that NadD strongly favors ATP and NaMN. On the other hand, NcdS-2 using CTP and NMN is about 45.4-fold and 213-fold more efficient than ATP and NaMN, respectively.

**Table 1 The kinetic parameters for wild-type NadD and NcdSs.**

| | Substrate | $K_m$ (×10⁻⁶ M) | $k_{cat}$ (×10⁻² s⁻¹) | $k_{cat}/K_m$ (×10³ M⁻¹ s⁻¹) | CTP preference[a] | Substrate | $K_m$ (×10⁻⁶ M) | $k_{cat}$ (×10⁻² s⁻¹) | $k_{cat}/K_m$ (M⁻¹ s⁻¹) | NMN preference[b] | Preference for NCD synthesis[g] |
|---|---|---|---|---|---|---|---|---|---|---|---|
| NadD | CTP[c] | 6069 | 1.72 | 2.83×10⁻³ | 1.84×10⁻⁵ | NMN[d] | 438 | 27.7 | 633 | 4.04×10⁻³ | 7.43×10⁻⁸ |
| | ATP[c] | 26.8 | 436 | 162 | | NaMN[d] | 32.4 | 506 | 156×10³ | | |
| NcdS-2 | CTP[e] | 7.50 | 29.6 | 39.4 | 45.4 | NMN[f] | 241 | 8.24 | 342 | 213 | 9.67×10³ |
| | ATP[e] | 42 | 3.65 | 0.87 | | NaMN[f] | 5380 | 0.86 | 1.61 | | |
| NcdS-3 | CTP[e] | 26.8 | 97.1 | 36.2 | 29.9 | NMN[f] | 226 | 13.9 | 612 | 12 | 3.59×10² |
| | ATP[e] | 33.9 | 412 | 1.21 | | NaMN[f] | 1830 | 9.32 | 50.9 | | |

$^a$CTP preference = $\frac{k_{cat}/k_m(CTP)}{k_{cat}/k_m(ATP)}$.
$^b$NMN preference = $\frac{k_{cat}/k_m(NMN)}{k_{cat}/k_m(NaMN)}$.
$^c$NaMN fixed.
$^d$ATP fixed.
$^e$NMN fixed.
$^f$CTP fixed.
$^g$Preference for NCD synthesis is defined as the product of CTP preference and NMN preference.

Similarly, NcdS-3 using CTP and NMN is 29.9-fold and 12-fold more efficient than ATP and NaMN, respectively. As NcdS was created in a semirational way by combining favorable mutations located in NTP-binding and pyridine nucleotide-binding pockets of NadD[19], the facts that both NcdSs reserved substrate preferences to CTP and NMN indicated that the two-substrate pockets were relatively separated. In total, NcdS-2 and NcdS-3 achieved $1.3 \times 10^{11}$-fold and $4.8 \times 10^{9}$-fold, respectively, higher preference for NCD synthesis compared with NadD. It was noteworthy that NcdS-2 and NcdS-3 had $K$m values for NMN at about 200 μM, which were around tenfold lower than those for NaMN, further suggesting these mutants competent to acquire NMN for NCD biosynthesis in vivo. As NcdS-2 showed lower $K$m for CTP and higher preference of NCD synthesis, it was selected for most of the subsequent experiments.

**Structural basis for NcdS.** To understand the molecular basis of the engineered NcdS, we crystallized NcdS-2 (PDB ID: 6KH2) and resolved the structure (Supplementary Fig. 5 and Supplementary Table 1). The resolution of the hexamer NcdS-2 was 2.6 Å. Compared with wild-type NadD (PDB ID: 1K4M), the loop 41–47 and the loop 133–144 had changed conformation, but the overall secondary structure and three-dimensional structure remained the same.

We have docked ATP and CTP into both NadD and NcdS-2, individually. It was known that ATP forms hydrogen bonds primarily with the main-chain atom of NadD, rather than side chain of amino acids[19]. In our docking results, adenine ring forms hydrogen bonds with the main-chain atoms of G107, F177, and I179 (Fig. 3a), whereas only one hydrogen bond between CTP and NadD is noticed (Fig. 3b). However, in NcdS-2, the P22K mutation leads to significantly reduced binding pocket for NTP. Moreover, all those hydrogen bonding interactions found in ATP–NadD complex are missing in the ATP–NcdS-2 complex, and only a weak hydrogen bond is observed between ATP and the side chain of R134 (Fig. 3c). In contrast, in the CTP–NcdS-2 complex the amino group of cytosine forms hydrogen bond with the oxygen atom of F177, and K22 forms additional hydrogen bond with the carbonyl oxygen in cytosine (Fig. 3d). Thus, hydrogen bonding network and steric hindrance of the NTP-binding pocket together contributed to CTP preference by NcdS-2.

In terms of the specificity of the pyridine nucleotide, H45 and W117 in NadD offer π–π stacking interactions with the nicotine ring of N(a)MN, and Y118 forms hydrogen bond with the carboxyl group and the amide group of NaMN and NMN, respectively (Fig. 3e, f). Therefore, NadD can accept both NaMN and NMN. For NcdS-2, no such π–π stacking interactions were noticed (Fig. 3g, h). The key residue D118 forms strong hydrogen bond with the amide group of NMN, while the interaction between D118 and the carboxyl group of NaMN is significantly reduced due to electrical repulsion. These structural features gave perfect explanation for the fact that NcdS-2 favors NMN over NaMN.

**Designing NCD self-sufficient E. coli strains.** To demonstrate in vivo NCD biosynthesis from CTP and NMN (Fig. 4a), we first engineered E. coli DH5α by overexpressing NcdS-2 under the control of a constitutive promoter gntT105P. We extracted cellular pyridine nucleotide cofactors and analyzed by high-performance liquid chromatography (HPLC). The engineered strain XYC2002 produced NCD at around 50 μM, while the parental strain DH5α failed to produce NCD (Fig. 4b). To further improve NCD production, we considered engineering efforts to increase precursor supply. It was reported that FtNadE originated

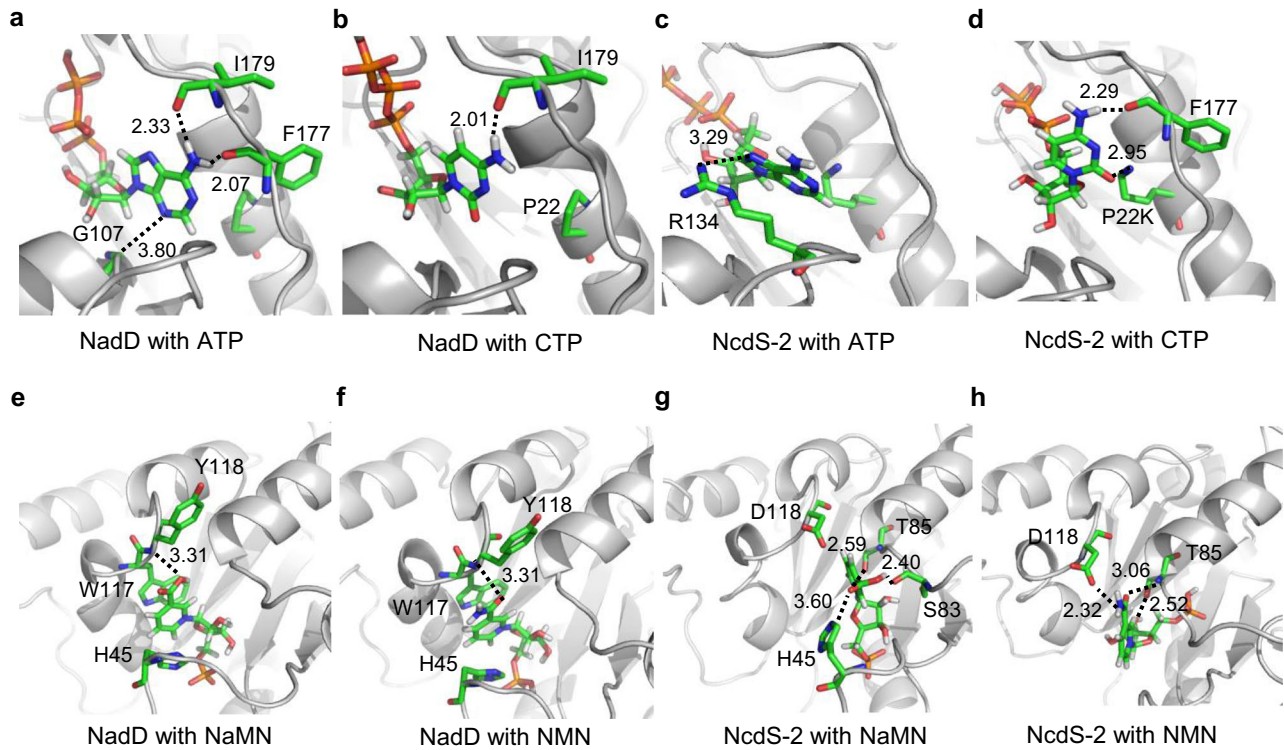

**Fig. 3 Docking analysis for substrate in wild-type NadD and NcdS-2.** The docking poses for ATP (**a**) and CTP (**b**) with NadD. The docking poses for ATP (**c**) and CTP (**d**) with NcdS-2. The docking poses for NaMN (**e**) and NMN (**f**) with NadD. The docking poses for NaMN (**g**) and NMN (**h**) with NcdS-2. The protein was shown as cartoon mode and in light gray. The substrate and the key residues were shown as stick mode. Green, carbon atoms. Blue, nitrogen. Gray, hydrogen. Red, oxygen. The dashed line indicates the hydrogen bond between ligand and receptor. The number marked near the dashed line represents the distance between two atoms (unit, angstrom).

from *Francisella tularensis* could catalyze the amidation of NaMN to NMN[22], and the CTP synthetase mutant *Ct*CTPS* (*Ct*CTPS-D149E) derived from *Chlamydia trachomatis* could produce CTP by UTP amidation[23]. Thus, we recruited *Ft*NadE and *Ct*CTPS* to boost the fluxes of the two precursors. When the three synthetases were coexpressed constitutively (Supplementary Fig. 6a), cellular NCD level of strain XYC2013 increased to 480 µM, while NAD level decreased substantially (Fig. 4b). These engineered cells grew noticeably slower, indicating that enhancing the fluxes of NMN and CTP led to unexpected effects in terms of NAD homeostasis and cell proliferation. To confirm the chemical identity of NCD found in the cell extracts, samples were also analyzed by using LC–MS/MS method and NCD molecular ions with expected fragmentation pattern were noticed (Fig. 4c, Supplementary Table 2). Moreover, no NCD was detected from another two strains without the overexpression of NcdS (Supplementary Fig. 7), indicating that NCD may indeed be considered as a non-natural redox cofactor in *E. coli.*

To further optimize NCD production, we engineered *E. coli* DH5α with a series of plasmids harboring different expression cassettes under the control of the arabinose operon (Supplementary Fig. 6b). Results showed that additional expression of either *Ft*NadE or *Ct*CTPS* was helpful for NCD synthesis (Fig. 4d). Interestingly, the highest level of NCD (5.0 mM) was achieved upon expression of all three synthetases in strain XYC2008, which was even higher than NAD level under normal cell growth conditions[18]. Again, NCD production led to reduced NAD levels.

The above results encouraged us assembling *NcdS*, *FtNadE*, and *Ct*CTPS* into the NCD biosynthesis module for more convenient strain engineering. We integrated one copy of the NCD biosynthesis module into the genome at the *arsB* loci in *E. coli* BW25113 (△*ldhA, dld::cat*) to give XYC5016. We also

replaced the high copy-replicon pMB1 ori of the plasmid pUC-WXY35 with a lower copy-replicon P15A, such that *E. coli* XYC5017 had fewer copies of the NCD biosynthesis module than XYC5008. Results showed that higher NCD concentrations were found when more copies of the NCD biosynthesis module were presented (Fig. 4e), thus suggesting a practical way to regulate cellular NCD levels. Furthermore, when the low-copy NCD biosynthesis module was transformed into DH5α and DH10B, the engineered strain XYC2017 and XYC8017 also produced NCD at 517 and 1776 µM, respectively (Fig. 4e), confirming the functional compatibility of the NCD biosynthesis module with different hosts.

**D-lactate production by NCD self-sufficient cells**. To demonstrate the usefulness of NCD self-sufficient cells, we tested their capacity to execute a metabolic circuit involving NCD-linked malic enzyme (Mae) mutant Mae*[10] and D-lactate dehydrogenase (Ldh) mutant Ldh*, the V152R/N213C mutant of wild-type Ldh (PDB code: 2DLD) originated from *Lactobacillus helveticus* (Fig. 5a). The NCD preference of Ldh* was estimated according to procedures in the recent study[14] as 7.51, which was 159-fold improvement over that of the wild-type enzyme. The rationale of the circuit is based on malolactic fermentation commonly used in winemaking process[24], where L-malate serves as both carbon source and reducing power donor. The metabolic circuit was assembled on plasmid pUC18 with Ldh* and Mae* coexpression controlled under the lac operon, and transformed into BW25113 (△*ldhA, dld::cat*) and XYC5016, namely BW25113 (Δ*ldhA, Δdld, arsB::Para-FtNadE-c-his-NcdS-2-CtCTPS**), to give strain XYD046 and XYD042, respectively. It should be noted that endogenous genes *ldhA* and *dld* were deleted in these host strains

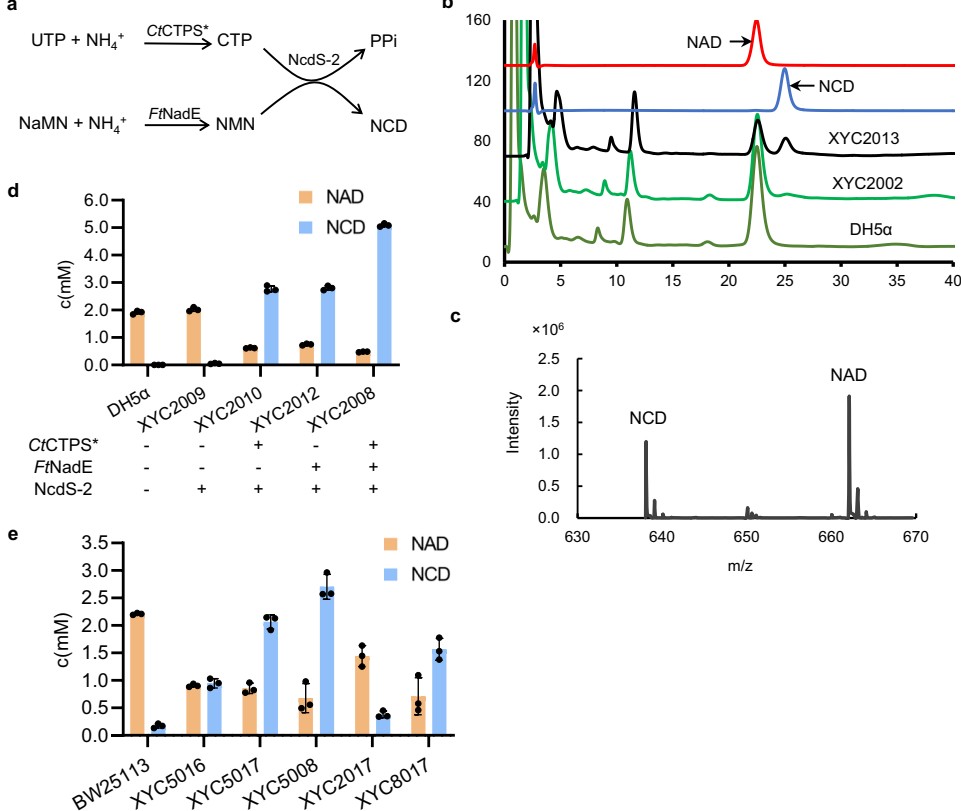

**Fig. 4 Construction of NCD self-sufficient *E. coli* strains. a** The designed NCD biosynthesis pathway. *Ft*NadE, *Ct*CTPS*, and NcdS-2 are used for biosynthesis of NMN, CTP, and NCD, respectively. **b** HPLC profiles of cofactors in DH5α and engineered strains. XYC2002, DH5α with constitutive expression of NcdS-2; XYC2013, DH5α with constitutive expression of *Ft*NadE, *Ct*CTPS* and NcdS-2. **c** The MS spectra of intracellular NAD and NCD in XYC2013. Negative ion mode. Experimental monoisotopic mass, NAD, 662.0941; NCD, 638.0850. Calculated molecular mass, NAD, 662.1018; NCD, 638.0906. Mass error, NAD, 11.6 ppm; NCD, 8.7 ppm. **d** Cofactor concentrations in DH5α strains with inducible expression of different enzymes. **e** Cofactor concentrations of engineered strains harboring different copy of the inducible NCD biosynthesis module. XYC5016, BW25113(Δ*ldhA, dld::cat*) with single copy of NCD biosynthesis module; XYC5017, BW25113(Δ*ldhA, dld::cat*) with the NCD biosynthesis module in low-copy plasmid; XYC5008, BW25113(Δ*ldhA, dld::cat*) with the NCD biosynthesis module in high copy plasmid; XYC2017, DH5α with the NCD biosynthesis module in low-copy plasmid; XYC8017, DH10B with the NCD biosynthesis module in low-copy plasmid. Experiments were conducted in triplicates ($n = 3$), data are presented as mean values ± SD. The experiments were performed independently at least twice, with similar results. Source data are available in the Source Data file.

such that Ldh* activity was solely responsible for D-lactate formation. Analysis of intracellular cofactor level indicated that XYD042 cells had 0.43 mM NAD and 0.27 mM NCD (Fig. 5b), both of which were lower than those found in the parental strain. We also detected the NCD- and NAD-linked activities of both Mae and Ldh in the crude extracts. While high NCD-linked activities were observed, background NAD-linked activities were also noticeable (Fig. 5c). The NAD-linked Mae activity may come from endogenous wild-type enzymes, and LDH activity from leaked activity by LDH*. When feed with L-malate, XYD042 cells consumed 1.20 mM L-malate and accumulated 0.65 mM D-lactate (Fig. 5d), indicating that 53.8% reducing power from L-malate was transferred to D-lactate. In comparison, XYD046 consumed 2.02 mM L-malate but accumulated 0.26 mM D-lactate, suggesting that only 13.0% reducing power from L-malate was used for D-lactate production. These results suggested that the NCD self-sufficiency indeed facilitated the expected function of the NCD-linked metabolic circuit.

## Discussion

Here, we created NcdS by reprograming the substrate-binding pockets of *E. coli* NaMNAT to use CTP and NMN as favorable substrates. Kinetic and structural analysis of these NcdSs afforded insights into the molecular basis of substrate preference switch.

We demonstrated that overexpression of NcdS-2 alone in *E. coli* enabled NCD biosynthesis, and higher NCD levels were achievable upon enhancing precursor supply. By coexpression of NcdS-2, *Ft*NadE, and *Ct*CTPS*, it was effective to engineer stable NCD self-sufficient phenotype. More significantly, the NCD self-sufficiency was shown to warrant an NCD-linked metabolic circuit orthogonal to the native NAD-linked metabolism in terms of cofactor dependence. Considering that nature has selected NADH over billions of years as indispensable cofactors particularly for redox metabolism, this study now expanded our capacity to use NCD(H) as functional cofactors in vivo. Our early studies used NCD transporter, chemical synthesized NCD, and NCD-linked enzymes to establish orthogonal reduction driving by phosphite or formate[11,13,25], the access of NCD self-sufficient hosts will greatly promote similar efforts.

This work offers a paradigm to design biosynthesis of NAD analogs with similar structural features. Except for ATP and CTP, other NTPs such as GTP, TTP, and UTP may be considered to attain nicotinamide guanosine dinucleotide (NGD), nicotinamide thymidine dinucleotide, and nicotinamide uridine dinucleotide. In fact, NMNAT3 in mammals is naturally active in synthesizing NGD from GTP in vitro[26] and NGD level was found similar to that of NAD in NMNAT3-expressing Tg mice[27]. These NAD analogs may be explored as noncanonical cofactors for devising

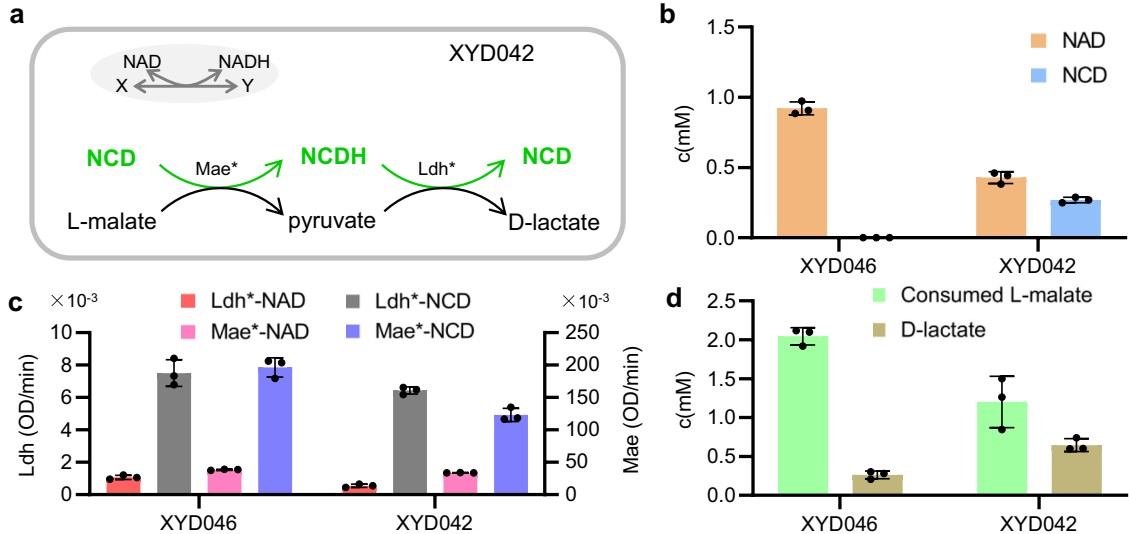

**Fig. 5 Application of NCD self-sufficient cells. a** NCD self-sufficient cells implanted with an NCD-linked metabolic circuit converting L-malate into D-lactate. **b** Intracellular cofactor concentrations. **c** Crude enzyme activities of Mae and Ldh with different cofactors. Ldh*-NAD assayed with NAD and D-lactate, Ldh*-NCD with NCD and D-lactate, Mae*-NAD with NAD and L-malate, Mae*-NCD with NCD and L-malate. **d** Profiles of L-malate consumption and D-lactate production. About $1.0 \times 10^{10}$ cells were incubated in MOPS buffer (pH 7.5) in the presence of 10 mM L-malate at 37 °C for 4 h. XYD042, BW25113 (ΔldhA, Δdld, arsB::Para-FtNadE-c-his-NcdS-2-CtCTPS*)/pUC-Plac-Ldh*-rbs-Mae*; XYD046, BW25113 (ΔldhA, dld::cat)/pUC-Plac-Ldh*-rbs-Mae*. Data are presented as mean values ± SD, error bars represent the standard deviation of data from three different cultures. The experiments were performed independently twice, with similar results. Source data are available in the Source Data file.

more orthogonal redox biochemistry in vivo and as probes for studying NAD-related nonredox processes[4].

Taken together, NcdS and NCD-linked enzymes now open new opportunities to consider more orthogonal redox chemistry at the metabolic level[28]. This will be amenable to those on-going efforts using non-natural bases and non-natural amino acids to expand our capacity in terms of understanding and reprograming life[29].

## Methods

**Strains, plasmids, and reagents**. Bacterial strains and plasmids are listed in Supplementary Tables 3 and 4. *E. coli* DH10B was used for directed evolution. *E. coli* BW25113 (ΔldhA, dld::cat) was used as host for D-lactate production. These two strains and *E. coli* DH5α were used to study the intracellular cofactors. All *E. coli* strains were cultured at 37 °C for activation, and at 30 °C for protein expression. FastPfu DNA polymerase was purchased from Transgen (Shanghai, China), and PrimeSTAR HS DNA polymerase and restriction endonucleases were purchased from Takara (Dalian, China). NMN, NaMN, alcohol dehydrogenase (ADH II), PMS, NBT, phenazine ethosulfate (PES), methylthiazolyldiphenyl-tetrazolium bromide (MTT), and isopropyl-β-thiogalactoside were purchased from Sigma-Aldrich (St. Louis, MO, USA). ATP, CTP, kanamycin, ampicillin, and lysozyme were purchased from Sangon Biotech (Shanghai, China). NAD and agar were purchased from Dingguo (Beijing, China). The standard NCD was chemically synthesized as described previously[10]. Briefly, to anhydrous DMSO (30 ml) was added NMN (1.2 g), Ph3P (6.5 g), (PyS)2 (5.5 g), and 1-methylimidazole (7.0 ml), and the mixture was stirred at 26 °C for 15 min. Then, a solution of CMP (4.0 g) tri-n-butylamine salt 13 in DMF (20 ml) was added. The reaction was held at 26 °C for 40 min, then stopped by adding acetone (100 ml) to form precipitates. The precipitates were collected and washed three times with acetone. Purification was performed by anion exchange column chromatography using 201 × 4 type anion resin (HCO2⁻ form) and eluted with 50 mM HCO2H. Fractions were concentrated and further purified on a DEAE Sephadex G-25 column eluted with 10 mM NH4HCO3. Fractions were pooled, lyophilized to give NCD as a white solid (880 mg, yield 38%). Primers (Supplementary Table 5) were synthesized by Sangon Biotech (Shanghai, China).

**Genetic methods**. To construct double- and triple-sites mutation libraries, PCR-based methods[30] were used to introduce mutations in the template plasmid pUC-kan-NadD expressing wild-type NadD with His₆ tag. Degenerate codons NNK (forward, N: adenine/cytosine/guanine/thymine; K: guanine/thymine) and NMN (reverse, M: adenine/cytosine) were utilized to construct saturated mutant libraries. A mixture of three kinds of codons CKC, TWT, and SAA (at a ratio of 1:1:1, W: thymine/adenine, S: guanine/cytosine) was used as forward primers, while GMG,

AWA, and TTS mixtures were used as reverse primers to introduce amino acids with large side chains (Arg, Leu, Phe, Tyr, Glu, and Gln) at defined sites. The mutant fragments amplified by the degenerate primer pairs were cloned into pUC-kan-NadD. The PCR products were transformed into electrocompetent DH10B cells after digesting with *Dpn*I.

To construct plasmids for NcdS overexpression, RF cloning strategy[31] was used and the resultant vectors pUC-kan-NcdS-2 and pUC-kan-NcdS-3 were obtained (Supplementary Fig. 3). For constitutive expression of NcdS-2, the coding sequencing was amplified from pUC-kan-NcdS-2 and inserted into pBCTB to generate pBCTB-NcdS-2.

For constitutive expression of *Ft*NadE, the coding sequence was amplified from pODC29-*Ft*NadE and inserted into pBCTA to generate pBCTA-*Ft*NadE. For constitutive expression of *Ct*CTPS*, the codon-optimized sequencing was amplified from pUC57-*Ct*CTPS* and then inserted into pCF to generate pUC-chl-PpykF-*Ct*CTPS*. The expressing cassettes of *Ct*CTPS* and NcdS-2 were assembled within the *Sac* I, *Xho* I, and *Nhe* I sites of pBCTA-*Ft*NadE, to afford the plasmid pBCT-WXY9, which harbors a constitutive NCD biosynthesis module (Supplementary Fig. 6a and Supplementary Table 6).

PCR-based strategy was used to construct the inducible NCD biosynthesis module (Supplementary Table 7). Primers pA-PolB-294 and pA-DedA-228 were used to amplify the *ara* operon from *E. coli* genome, and the fragment was inserted into pUC18 to generate pUC-PolB-DedA. The original genes *ara*B, *ara*A, and *ara*D were replaced by the coding sequence of *Ft*NadE, NcdS-2, and *Ct*CTPS*, respectively (Supplementary Fig. 6b), according to the RF cloning strategy with PCR generated megaprimers[31].

For the NCD(H) utilization and regeneration vector, plasmid pK-Ldh-V152R/N213C was used as templates for NCD-dependent Ldh*, while plasmid pK-Mae*-Pdh* was used the backbone of double-gene coexpression vector. All genes harbored a His₆ tag at the C-terminal. Then, we amplified the ribosome binding site from pET-24b and linked the open reading frames of the two genes. The PCR products digesting after *Dpn*I were transformed into BW25113 (ΔldhA, dld::cat) and XYC5016, respectively.

**High-throughput screening for NCD biosynthesis activity**. The procedures of preparing crude enzyme extracts and the high-throughput screening strategy were established as described previously[18]. Briefly, colonies from each library were cultured in 96-well deep plates for protein expression. Cell pellets were harvested by centrifugation and lysed by lysis buffer containing lysozyme (1 mg/ml). The supernatants after centrifugation were taking as crude enzyme extracts. A volume of 50 μl substrate-mixture containing 50 mM HEPES (pH 7.5), 5.0 mM MnCl₂, 10 mM MgCl₂, 2.0 mM NMN, 2.0 mM CTP, and 10 μl crude enzyme extracts was shaken at 37 °C, 200 rpm for 2 h. Then, a volume of 10 μl dyeing-mixture containing 50 mM HEPES (pH 7.5), 5.0 mM MnCl₂, 10 mM MgCl₂, 30 mM L-malate, 0.15 mM PMS, 0.6 mM NBT, and 0.06 U purified Mae* (Mae-L310R/Q401C) was added and incubated at room temperature for 1 h.

Variants of those turned blue in the initial screening experiments were selected for crude enzyme activities assay. The detection method was as described above

with minor modifications. Briefly, the mutant strains were cultured in 2.5 ml of LB in 24-well deep plates. Crude enzyme extracts were prepared and exact 100 µl of the extracts was added into 300 µl substrate-mixture containing 0.1 mM NMN and 2.0 mM ATP or CTP. The mixture was incubated at 37 °C for 2 h, quenched by adding an equal volume of chloroform, and then centrifuged at 14,200 × g for 15 min. A 45 µl of the supernatant was transferred into dyeing-mixture containing 50 mM HEPES (pH 7.5), 5 mM MnCl$_2$, 10 mM MgCl$_2$, 5.0 mM L-malate, 0.4 mM PES, 1.0 mM MTT, and 0.8 U purified Mae or Mae* in a total volume of 100 µl. The absorbance at 570 nm was monitored at room temperature with PowerWave XS universal microplate spectrophotometer (BioTek Instruments Inc., USA). Data were analyzed by KC JrWin (v1.41.6). Plasmids from the positive hits were extracted and sequenced by Sangon Biotech (Shanghai, China).

**Specific enzyme activity and kinetic assays**. All specific activity and kinetic analysis were done with purified proteins by using enzyme-coupled methods. Standard reaction mixtures (150 µl) contained 50 mM HEPES (pH 7.5), 5.0 mM MnCl$_2$, 10 mM MgCl$_2$, 0.1 mM NMN, 2.0 mM CTP or ATP, 10 mM L-malate, 1.0 U purified Mae* or Mae, and 0.1–0.5 mg NadD mutant. The absorbance at 340 nm was monitored at 25 °C with Thermo Scientific Evolution 220 UV-Visible Spectrophotometer (BioTek Instruments Inc., USA). Data were analyzed by Thermo InSight (v2.0.459).

To determine kinetic parameters with different substrates, procedures were as follows. Enzymatic activities toward NMN and NaMN of NadD were measured using ATP as cosubstrate as described previously with minor modification[18]. Activities toward NMN and NaMN of NcdS were measured as follows. A volume of 100 µl assay mixture contained 50 mM HEPES (pH 7.5), 4.0 mM CTP, 10 mM MgCl$_2$, 5.0 mM MnCl$_2$, 10 mM L-malate, 10 U Mae*, 1.0 mM PES, and 0.4 mM MTT. The concentration of NMN varied from 50 to 5000 µM, and the concentration of NaMN varied from 100 to 10,000 µM. Activities toward CTP and ATP were measured by coupling with Mae* or alcohol dehydrogenase. For CTP activity, a volume of 100 µl assay mixture contained 50 mM HEPES (pH 7.5), 4.0 mM NMN or NaMN, 10 mM MgCl$_2$, 5.0 mM MnCl$_2$, 10 mM L-malate, 10 U Mae*, 1.0 mM PES, and 0.4 mM MTT. For ATP activity, 100 µl of assay mixture contained 50 mM HEPES (pH 7.5), 4.0 mM NMN or NaMN, 10 mM MgCl$_2$, 45.8 mM ethanol, 5.0 U alcohol dehydrogenase, 1.0 mM PES, and 0.4 mM MTT. The concentration of CTP varied from 500 to 15,000 µM, and the concentration of ATP varied from 5 to 4000 µM. The reaction was initiated by adding appropriate amount of purified enzyme, and the absorbance at 570 nm was monitored at room temperature with PowerWave XS universal microplate spectrophotometer (BioTek Instruments Inc., USA). Data were analyzed by KC JrWin (v1.41.6). The kinetic data were determined from a Lineweaver-Burk plot.

**X-ray structural analysis**. The three-dimensional structure of wild-type NadD (PDB ID: 1K4M) was obtained from PDB server (http://www.rcsb.org/pdb/home/home.do). The crystal of mutant NcdS-2 was grown in solution containing 1.4 M sodium phosphate monobasic monohydrate–potassium phosphate dibasic (pH 8.0) by sitting-drop vapor diffusion method at 16 °C. The crystal was flash-cooled in a nitrogen stream. Diffraction data were collected at beam line BL18U1 of the Shanghai Synchrotron Radiation Facility at the walvelength of 1.0001 Å. Corresponding parameters were determined using the program HKL2000 (v715-linux-x86_64)[32], and crystal was solved by molecular replacement method using the PHASER program in PHENIX (v1.13_2998)[33]. Manually and automated refinement were performed with COOT (v0.8.9) and PHENIX, respectively. The statistical score of the Ramachandran plot shows that 96.21% atoms are in the most favored regions, and additional 3.46% in the allowed regions. All structural figures were prepared using Pymol (0.99rc6).

**Molecular docking**. Molecular docking for interaction analysis was carried out using the MOE (v2018.01) package (Chemical Computing Group, Montreal, Canada). For each substrate-enzyme pair, we choose the conformation with lowest conformation energy and best similarity with natural NAD–NadD complex (PDB ID: 1K4M).

**Intracellular cofactor analysis**. The procedures of protein induction and cofactor extraction were carried out as described with minor modifications[34]. The engineered strains were inoculated in 5 ml LB media with proper antibiotics at 37 °C for 14 h. The seed culture was then inoculated into a fresh LB media with ampicillin at 30 °C for 24 h at an initial OD$_{600}$ of 0.05. About 2.0 × 10$^{10}$ cells were collected by centrifugation at 14,200 × g for 5 min at 4 °C. The pellet was resuspended in 500 µl extraction solution of acetonitrile, methanol, and water (40:40:10, vol:vol:vol) prechilling to −20 °C, vortexed, and then incubated at −20 °C for 15 min. The extraction procedure was repeated thrice. The supernatant in each cycle was mixed together and stored at −80 °C for further analysis within 3 days.

Cofactor extracts were analyzed at 25 °C by using an Acchrom HPLC system (Beijing, China) incorporated with a Luna Aminopropyl column (150 mm × 1.0 mm, 3 µm, Phenomenex, USA). The elution was at flow rate of 0.08 ml/min with an isocratic mobile phase containing 71.5% acetonitrile, 3.0 mM NH$_4$COOH, and 6.0 mM NH$_4$OH. The detection was held for 30 min for each sample and the sample injection volume was 10 µl. Data were analyzed by Empower3 (v2010.3471-c).

LC–MS/MS analysis was performed on an ACQUITY UPLC System (Waters Corp., Milford, USA) hyphened to an AB Sciex Qtrap 5500 instrument (AB Sciex Pte. Ltd, Foster City, CA). The cofactor extracts were firstly identified by comparison of their retention times and MS/MS spectra with standards. The separation was carried out on the same Luna Aminopropyl column maintained at 30 °C. The isocratic mobile phase was 52.5% acetonitrile with 5.0 mM NH$_4$COOH. The injection volume was 5 µL and the flow rate was 0.2 ml/min. ESI ion source operated in negative MRM mode and nitrogen was the only gas used. The ion spray voltage was set at −4500 V, source temperature at 500 °C, curtain gas (CUR) at 25 psi, nebulizer gas (GS1) at 40 psi, and desolvation gas (GS2) at 60 psi. Collision gas (CAD) was set at medium. Compound-related MS parameters are listed in Supplementary Table 2. The cofactor standards were dissolved in extraction solution (methanol:acetonitrile:H$_2$O, 40:40:20, pH was adjusted by NH$_4$OH to 8.5). Data were analyzed by software Analyst (v1.6).

**Procedures for D-lactate production with NCD self-sufficient strains**. To demonstrate the conversion of L-malate into D-lactate, about 1.0 × 10$^{10}$ cells were collected by centrifugation at 14,200 × g at 4 °C for 5 min and washed with 1 ml of 50 mM MOPS buffer (pH 7.5). The cells were resuspended in 500 µl of reaction solution containing 50 mM MOPS (pH 7.5) and 10 mM L-malate and the reaction was held at 37 °C, 200 rpm for 4 h. Reaction was quenched by taking 180 µl of reaction solution into the mixture of acetonitrile, methanol, and water (40:40:10, vol:vol:vol, prechilled at −20 °C), and the metabolites were extracted by holding at −20 °C for 20 min. The supernatants were collected by centrifugation at 14,200 × g at 4 °C for 5 min, and stored at −80 °C. Both L-malate and D-lactate were determined by enzyme-coupled assays. The reaction was monitored on a Synergy H1 microplate reader at 570 nm, and standard cures were drawn using the corresponding standards of different concentrations. Data were analyzed by Gen5 CHS (v3.00).

To analyze crude enzyme activities of those cells for D-lactate production, about 1.0 × 10$^{10}$ of cells were collected and resuspended in 400 µl of cell lysis solution (100 mM Tris-HCl, 1.0 mM MgCl$_2$, 1.0 mg/ml lysozyme, and 0.1 mg/ml DNase) at 37 °C for 4 h. Mae* and Ldh* activities were measured in a mixture containing 50 mM HEPES (pH 7.5), 10 mM MgCl$_2$, 0.05 mM NAD or NCD, 0.4 mM MTT, 1.0 mM PES, and 5.0 mM L-malate for Mae* or D-lactate for Ldh*. The reaction was started by addition of 15 µl of cell lysates. Reaction rates were obtained by monitoring the absorbance at 570 nm at room temperature with PowerWave XS universal microplate spectrophotometer. Data were analyzed by KC JrWin (v1.41.6).

To determine cellular cofactor levels in those cells for D-lactate production, cofactors were extracted as described in ref. [35]. Briefly, 5.0 × 10$^9$ cells were collected by centrifugation at 8000 × g for 2 min at 4 °C, and the cell pellets were washed with phosphate-buffered saline buffer. About 250 µl of 0.2 M HCl was added to the cell pellets, which were incubated at 55 °C for 10 min. Then, the cells were cooled on ice for 5 min. About 250 µl of 0.1 M NaOH was added to the extracts for neutralization. The supernatant after centrifugation was stored at −20 °C for further measurement. NAD was quantified by an ADH-coupled assay[35]. NCD was quantified by a Mae*-coupled colorimetric assay in 50 mM HEPES (pH 7.5) containing 10 mM MgCl$_2$, 5.0 mM MnCl$_2$, 1.0 mM PES, 0.4 mM MTT, 5.0 mM L-malate, and 3.0 U Mae*. The reaction was started by adding 15 µl NCD sample and the absorbance at 570 nm was monitored at room temperature with PowerWave XS universal microplate spectrophotometer. Data were analyzed by KC JrWin (v1.41.6).

**Reporting summary**. Further information on research design is available in the Nature Research Reporting Summary linked to this article.

## Data availability
Coordinates and structure data of NcdS-2 have been deposited in the Protein Data Bank with the accession code 6KH2. Coordinates and structure data of EcNadD are available in the Protein Data Bank with the accession code 1K4M. The gene and protein used in this study are available in NCBI with the accession codes as follows: EcNadD, QJZ11375; FtNadE, EDZ90683; CtCTPS, AAA80195; EcMae, NP415996; and LhLdh, WP003628108. Source data are provided with this paper. All other data are available from the corresponding author upon request.

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

## Acknowledgements

The authors thank Prof. Andrei L. Osterman (Burnham Institute for Medical Research, USA) for providing pODC29-*Ft*NadE. The authors thank Li Wang (Dalian Institute of Chemical Physics, CAS, China) for helping analyze LC/MS data, Dr. Yuxue Liu for determination of NCD preference of Ldh*, and Energy Biotechnology Platform of Dalian Institute of Chemical Physics for providing advanced facilities. This work was supported by National Key R&D Program of China (No. 2019YFA0904900), National Natural Science Foundation of China (Nos 21778053, 21907092, 21837002, 21721004), and Dalian Institute of Chemical Physics, CAS (Nos DICP BioChE-X201801, DICP I202020).

## Author contributions

Z.K.Z. conceived the project and supervised the research. X.W. performed most experiments and drafted the manuscript. Y.F. resolved the crystal structure of NcdS-2. X.G. constructed the target fragment of the NCD biosynthesis module for genome editing. Q.W. analyzed the intracellular metabolites by LC/MS. S.N., Q.L., and J.W. helped in preparation of materials and reagents. L.W. constructed BW25113 (△*ldhA, dld:: cat*) and engineered the NCD-preferred Ldh*. Z.K.Z. and X.W. revised the manuscript. All authors contributed to discussion of results and comment on this paper.

## Competing interests

The authors declare no competing interests.
