## [Peer Review File · Nature Communications]

Reviewers' Comments:

Reviewer #1:

Remarks to the Author:

The key results of this work are the development of enzymes for biosynthesis of the non-natural cofactor nicotinamide cytosine dinucleotide (NCD) and NCD self-sufficient *E. coli* strains. The authors created some engineered enzymes that can be designated as NCD synthetases by reprogramming the substrate binding pockets of ATP and nicotinic acid mononucleotide (NaMN) of *E. coli* NaMN adenylyltransferase (NaMNAT) to use CTP and nicotinamide mononucleotide (NMN) as favorable substrates. They also elucidated the molecular basis of the substrate preference switch of these NCD synthetases by kinetic and structural analysis. Furthermore, they constructed a stably NCD self-sufficient *E. coli* strain by co-expressing the NcdS with several enzymes that enhance precursor supply, and demonstrated the applicability of the NCD self-sufficient cells by D-lactate production test using NCD-linked malic enzyme and D-lactate dehydrogenase mutants. The novel insights in NCD-linked redox chemistry *in vivo* provided in this study are interesting and deserve to be considered for publication in *Nature Communications*. However, some comments should be considered before the publication.

Specific comments:

1. Line 125. The definition of "preference for NCD synthesis" is unclear. The authors should clarify the definition.
2. Line 127. To conclude that the substrate binding pockets of the NcdSs are relatively independent, a comparison of the kinetic parameters with enzymes in which ATP- and NaMN-binding pockets are mutated individually would be necessary.
3. Line 132. The authors should explain why they chose NcdS-2 over Ncd-3 for subsequent experiments.
4. Line 191. Specify appropriate reference(s) for NCD-linked D-lactate dehydrogenase mutant Ldh*.

Reviewer #2:

Remarks to the Author:

In the work by Wang and coworkers the authors created engineered enzymes and genetic/enzyme circuits to produce large amounts NCD a non-natural redox cofactor using *E. coli* as host. They start with the construction of multiple-site mutation libraries which allowed to screen NadD variants with altered substrate preference. They identified and characterized kinetically and at structural level NadD variants which changed the preference of the natural substrates ATP and NaMN to CTP and NMN in such way that cells carrying these enzyme variants were able to produce the non-natural redox cofactor NCD. The authors were able to increase cellular NCD production by performing elegant experiments where the availability of the substrates CTP and NMN were increased using additional enzymes which directed the biosynthesis of these substrates. Furthermore, the authors showed that it is possible to use the non-natural NCD to drive specialized reactions by applying additional engineered enzymes which can use NCD instead of NAD as substrate. This study provides an important base for future biotechnology engineering studies. The paper is easy to understand despite the complexity of some assays. The conclusions are well justified by the data presented.

Some comments that may be considered by the authors:

Page 7, line 116. They authors used coupled enzyme assays to perform kinetic analysis. This includes the use of Mae* to determine the formation of both NCD and NaCD. It would be important to indicate for the readers to what extent Mae* is also able to use NaCD as substrate and/or if

Mae* was in excess in the assay in such way that the kinetic parameters were not affected by the low reactivity of the coupled enzyme.

Page 8, line 158. The authors indicate that strain XYC2002 produced NCD and refer to figure 4b. However, it is impossible to observe the NCD peak in this strain in the referenced figure. If the peak was low intensity I suggest providing and insert graphic with a zoom in the referred NCD peak.

Page 9, line 169. It is difficult to understand how the isotopic distribution was used to confirm the molecular identity of the compounds (this is possible but not commonly applied). I think the authors are refereeing to the experimental determined monoisotopic mass (all C12) matching the expected calculated molecular mass to the compounds. Please make clear if that was the case in the text. I also suggest indicating the experimental mass determined in the assay (Fig 4b) an compared with the calculated and indicate mass error (if any) in Da or ppm (this can be included in the Fig 4 legend). In, Fig. S7 you must indicate which ion is been extracted (i.e m/z and the considered mass error)

Page 10, line 189. The text of this section can be further improved as it was difficult to follow. It would be good if the authors provide some indication of the strains genotype while they are cited in the text and in the figures, it is difficult to follow the track as many strains with different codes were used. Furthermore, in the current version the reader must see the info in two tables (strains and plasmids) to follow what is the genotype of each strain. I was a bit confused with fig 5c, first, I suggest using color coded bars to avoid confusion. Please indicate in the text and/or in the figure legend what is being measured in fig 5c. Is this in vitro measurement using crude extracts? If so which substrates were added to the extracts to measure each enzyme activity?

Fig 2b. This is difficult to follow and therefore its quality must be improved. I suggest removing the background surface density of the figure. Alternatively, the authors could show only a surface representation of the binding pocket in the main structure with NAD in sticks representation. Please also correct the legend (316) and the protein structure is not represented in cartoon mode.

Fig S1. Please provide information on the meaning of mOD? Please indicate in the legend if these assays were performed using crude extracts? Please indicate in the legend respective concentrations of the substrates used. Please indicate in the legend the difference between DH10B and wt?

Fig S2. Please indicate in the legend if these assays were performed using crude extracts? Please indicate in the legend respective concentrations of the substrates used.

Fig S3. This figure does not help, please use the legend to give details on how the construction was made or indicate what is happening in each step just beside in the down arrows used to connect each step.

Response to editors and reviewers for NCOMMS-21-02937

Dear Editors and Reviewers,

We appreciate your constructive comments and suggestions. Accordingly, we revised the manuscript. Our detailed responses are included below.

Reviewer #1

Comment 1. Line 125. The definition of "preference for NCD synthesis" is unclear. The authors should clarify the definition.

Reply: We have added the definition of "preference for NCD synthesis" in Table 1 in the revised manuscript. In short, preference for NCD synthesis of a given protein is defined as the product of CTP preference and NMN preference.

Comment 2. Line 127. To conclude that the substrate binding pockets of the NcdSs are relatively independent, a comparison of the kinetic parameters with enzymes in which ATP- and NaMN-binding pockets are mutated individually would be necessary.

Reply: Thanks for pointing out this issue. Indeed, more experiments should be done to make the original statement. According to the strategy used here, we basically combined favorable mutations located in NTP-binding and pyridine nucleotide-binding pockets of NadD to generate NcdS. Therefore, the facts that both NcdSs reserved substrate preferences to CTP and NMN indicated the two substrate pockets were relatively separated. We have made revisions to clarify this (Line 118-121).

Comment 3. Line 132. The authors should explain why they chose NcdS-2 over Ncd-3 for subsequent experiments.

Reply: We have added the explanation in the revised manuscript (Line 125-126). In short, as NcdS-2 had lower K_m for CTP and higher preference of NCD synthesis, it was selected for more detailed study.

Comment 4. Line 191. Specify appropriate reference(s) for NCD-linked D-lactate dehydrogenase mutant Ldh*.

Reply: We added detailed information for Ldh* in the revised manuscript (Line 184-185). In short, Ldh* is the V152R/N213C mutant of wild type Ldh (PDB code: 2DLN) originated from *Lactobacillus helveticus*. It was initially constructed by one of the co-author Lei Wang and has not been published yet.

Reviewer #2:

Comment 1. Page 7, line 116. The authors used coupled enzyme assays to perform kinetic analysis. This includes the use of Mae* to determine the formation of both NCD and NaCD. It would be important to indicate for the readers to what extent Mae* is also able to use NaCD as substrate and/or if Mae* was in excess in the assay in such way that the kinetic parameters were

not affected by the low reactivity of the coupled enzyme.

Reply: Thanks for the comment. In our enzyme-coupled activity assays, we tested different doses of the coupling enzyme to make sure that the coupling enzyme would not limit the target enzyme activity. We added the experimental information for kinetic analysis in the revised manuscript (line 292-303).

Comment 2. Page 8, line 158. The authors indicate that strain XYC2002 produced NCD and refer to figure 4b. However, it is impossible to observe the NCD peak in this strain in the referenced figure. If the peak was low intensity I suggest providing and insert graphic with a zoom in the referred NCD peak.

Reply: Thanks for the comment. The peak of NCD in strain XYC2002 was low intensity, and we have zoomed in the intensity coordinates of strain DH5 α , XYC2002 and XYC2013 for comparison in Fig. 4b in the revised manuscript.

Comment 3. Page 9, line 169. It is difficult to understand how the isotopic distribution was used to confirm the molecular identity of the compounds (this is possible but not commonly applied). I think the authors are refereeing to the experimental determined monoisotopic mass (all C12) matching the expected calculated molecular mass to the compounds. Please make clear if that was the case in the text. I also suggest indicating the experimental mass determined in the assay (Fig 4b) an compared with the calculated and indicate mass error (if any) in Da or ppm (this can be included in the Fig 4 legend). In, Fig. S7 you must indicate which ion is been extracted (i.e m/z and the considered mass error)

Reply: Thanks for the suggestions. We used chemically synthesized NCD and commercialized NAD as standards for LC-MS/MS analysis. The m/z of standard NCD and NAD are 638.0827 and 662.0936, respectively. On the other hand, we selected 638.000/516.000 and 662.000/540.000 as characteristic ion pairs for multiple reaction monitoring analysis of NCD and NAD, respectively. We have added the information in Fig 4 legend in the revised manuscript and Fig. S7 in Supporting information.

Comment 4. Page 10, line 189. The text of this section can be further improved as it was difficult to follow. It would be good if the authors provide some indication of the strains genotype while they are cited in the text and in the figures, it is difficult to follow the track as many strains with different codes were used. Furthermore, in the current version the reader must see the info in two tables (strains and plasmids) to follow what is the genotype of each strain. I was a bit confused with fig 5c, first, I suggest using color coded bars to avoid confusion. Please indicate in the text and/or in the figure legend what is being measured in fig 5c. Is this in vitro measurement using crude extracts? If so which substrates were added to the extracts to measure each enzyme activity?

Reply: Thanks for the comment. First, we have added the genotype information of engineered strain XYC5016 in the revised manuscript. Second, we have revised Table S3 with proper

plasmid information to make them easier to follow. In terms of Fig. 5c, activities were assayed with crude extracts. We added the information in the revised text (Line 193-194), revised the color bars and legend of Fig. 5c.

Comment 5. Fig 2b. This is difficult to follow and therefore its quality must be improved. I suggest removing the background surface density of the figure. Alternatively, the authors could show only a surface representation of the binding pocket in the main structure with NAD in sticks representation. Please also correct the legend (316) and the protein structure is not represented in cartoon mode.

Reply: Thanks for the comment. We have removed the background surface density of ATP-binding pocket in Fig. 2b and correct the legend in the revised manuscript.

Comment 6. Fig S1. Please provide information on the meaning of mOD? Please indicate in the legend if these assays were performed using crude extracts? Please indicate in the legend respective concentrations of the substrates used. Please indicate in the legend the difference between DH10B and wt?

Reply: Thanks for the comment. mOD is a unit in the software of the Bio-tek PowerWave XS universal microplate spectrophotometer, and it means 1×10^{-3} OD. We have change the unit mOD to 1×10^{-3} OD in Fig. 5c and Fig. S1. Also, assay information and difference between DH10B and WT were added in the legend of Fig. S1 during revision.

Comment 7. Fig S2. Please indicate in the legend if these assays were performed using crude extracts? Please indicate in the legend respective concentrations of the substrates used.

Reply: Thanks for the comment. For specific activity assay, purified proteins were used. We have added the information in Fig. S2 in the revised manuscript.

Comment 8. Fig S3. This figure does not help, please use the legend to give details on how the construction was made or indicate what is happening in each step just beside in the down arrows used to connect each step.

Reply: Thanks for the comment. We added the description of the process as legend Fig. S3 in the revised manuscript.

A revised manuscript with the correction sections red marked was attached as the supplemental material and for easy check/editing purpose.

Should you have any questions, please contact us without hesitate.

Reviewers' Comments:

Reviewer #1:

Remarks to the Author:

The manuscript improved greatly through revisions. However, the authors' explanation of LDH* is inadequate. If LDH* has not been published yet, they should provide experimental results showing that LDH* utilizes NCD as its preferred cofactor compared to *Lactobacillus helveticus* wild type Ldh.

Reviewer #2:

Remarks to the Author:

The authors have fully covered all concerns raised in the previous version of the paper.

Response to editors and reviewers for NCOMMS-21-02937

Dear Editors and Reviewers,

We appreciate your constructive comments and suggestions. Accordingly, we revised the manuscript. Our detailed responses are included below.

Reviewer #1

Comment 1. The manuscript improved greatly through revisions. However, the authors' explanation of LDH* is inadequate. If LDH* has not been published yet, they should provide experimental results showing that LDH* utilizes NCD as its preferred cofactor compared to *Lactobacillus helveticus* wild type Ldh.

Reply: Thanks for your comments. Indeed, the NCD preference of Ldh* was estimated previously as 7.51, which was 159-fold improvement over that of the wild type enzyme. The experiment was done by Dr. Yuxue Liu and the procedures could be found in Reference 14. We acknowledge Dr. Liu's contribution. The information was now included in the revised text (Line 190-192).

Editor's Comment

Your manuscript has been checked for clarity and against journal policies and formatting style. The issues listed below must be addressed; failure to do so will cause delays in acceptance.

Please highlight all changes in the manuscript text file, either using the track changes feature in Microsoft Word or coloured highlighting in LaTeX.

Please include your response to these requests in the space provided and return this checklist with your final submission.

Reply: We have further revised the manuscript according to formatting instructions and requests listed in the checklist.

A revised manuscript with the correction sections red marked was attached as the supplemental material and for easy check/editing purpose.

Should you have any questions, please contact us without hesitate.